# The Risk Prediction of Coronary Artery Lesions through the Novel Hematological Z-Values in 4 Chronological Age Subgroups of Kawasaki Disease

**DOI:** 10.3390/medicina56090466

**Published:** 2020-09-11

**Authors:** Hwa Jin Cho, Woo Young Kim, Sung Man Park, Jung Hwa Lee, Hong Ju Shin, Gi Young Jang, Kee Soo Ha

**Affiliations:** 1Department of Pediatrics, Chonnam National University Children’s Hospital & Medical School, #42, Jebong-ro, Dong-gu, Gwangju 61469, Korea; chhj98@gmail.com; 2Department of Surgery, Korea University Guro Hospital, #148, Gurodong-ro, Guro-gu, Seoul 08308, Korea; drwykim@gmail.com; 3Department of Pediatrics, Korea University Guro Hospital, #148, Gurodong-ro, Guro-gu, Seoul 08308, Korea; psm8550@hanmail.net (S.M.P.); leejmd@chollian.net (J.H.L.); 4Department of Thoracic and Cardiovascular Surgery, Korea University Ansan Hospital, #123 Jeokgeum-ro Danwon-gu, Ansan, Gyeonggi-do 15355, Korea; babymedi@naver.com; 5Department of Pediatrics, Korea University Ansan Hospital, #123 Jeokgeum-ro Danwon-gu, Ansan, Gyeonggi-do 15355, Korea; jgynhg@naver.com

**Keywords:** leukocyte count, hemoglobin, coronary artery disease, age distribution, Kawasaki disease

## Abstract

*Background and Objectives:* Most cases of Kawasaki disease (KD) occur between the ages of 6 months and 5 years. Differences in immunological reaction and CAL (coronary artery lesion) by the age subgroups classified according to the prevalence of KD and those particularly in the earlier life of KD should be investigated. *Materials and Methods:* The laboratory data of 223 infantile and 681 non-infantile KD cases from 2003 to 2018 at Korea University Hospital were retrospectively analyzed. Patients with KD were divided into infants and non-infants and further subdivided into four subgroups by age. The age-adjusted Z-values were compared among the subgroups. Febrile controls were identified as patients with fever for >5 days and who showed some of the KD symptoms. *Results:* IVIG (intravenous immunoglobulin) resistance at the age of 6 months or less was significantly lower than that at the ages of 7–12 months and 13–60 months (respectively, *p* < 0.05). The significant risk factors for CAL in total KD patients were age, incomplete KD, post-IVIG fever, IVIG resistance, convalescent Z-eosinophil, and subacute platelet (*p* < 0.05). The significant risk factors for CAL at the age of 6 months or less were IVIG resistance, acute Z-neutrophil, subacute Z-neutrophil, subacute NLR (neutrophil to lymphocyte ratio), and subacute platelet (respectively, *p* < 0.05). *Conclusion:* Younger age and incomplete presentation in KD can be independent risk factors for CAL. The immune reactions of KD at a younger age are more tolerated compared with those at older ages during the acute phase. The immune response at the age of 6 months or less showed immune tolerance in terms of incomplete presentation and IVIG responsiveness. The risk factors such as IVIG resistance, subacute platelet, subacute NLR, and acute or subacute Z-neutrophil at the age of 6 months or less can be very useful parameters to predict CAL in young, incomplete KD.

## 1. Introduction

Infants with Kawasaki disease (KD) in the first year of life (particularly in the first 6 months) have more frequent cardiac sequelae and greater morbidity than older children [1]. The high risk of coronary artery lesions (CAL) in infantile KD is associated with a high incidence of incomplete presentation [2]. Neonatal and infantile immunity is complex and distinct from the immunity of older children and adults, in that the neonatal and infantile periods represent a transitional phase from the internal uterine environment to the external microbe-laden world [3]. Immunological differences between infantile KD and non-infantile KD can lead to differences in pathogenesis; however, the exact mechanisms of illness in KD are yet to be revealed.

A previous study reported that most cases of KD (>80% of cases) occur between the ages of 6 months and 5 years, and the occurrence of KD before 6 months or after 5 years of age is rare [4]. Our previous investigation showed that the risk of CAL in infantile KD throughout all phases was highly associated with lower Z-hemoglobin and thrombocytosis compared with non-infantile KD [5]. It is necessary to investigate whether there are differences in immunological reaction and CAL by the age subgroups classified according to the prevalence of KD, and it is particularly important to detect latent CAL early through useful predictors in young infants with KD. In the present study, patients with KD were divided according to age cutoffs of 6 months (infants) and 5 years (non-infants). Age-adjusted Z-values were compared among age subgroups, and the hematological cell differences between each subgroup were investigated with respect to white blood cell (WBC) and red blood cell (RBC) counts.

## 2. Materials and Methods

### 2.1. Patients

We retrospectively investigated children with KD who received acute treatment and underwent echocardiographic and laboratory blood testing from 2003 to 2018 at Korea University Guro Hospital. In total, 223 patients with infantile KD and 681 patients with non-infantile KD were enrolled in this study. A total of 31 children aged <6 months and 14 children aged >5 years were included in the extreme age subgroup during an additional 3-year period (from 2016 to 2018), compared to our previous study [5] Children with KD were divided into the infantile and non-infantile KD groups and further subdivided by ages of 6 months and 5 years (4 subgroups) according to rare or predominant incidences. The extreme age subgroups were defined as extreme young (EY, <6 months) and extreme old (EO, >5 years). Typical age subgroups were defined as typical young (TY, 7–12 months) and typical old (TO, 13–59 months).

Children were classified as febrile controls (FCs) if they had fever for >5 days and showed some of the KD symptoms. The FCs were initially suspected of having incomplete KD because of prolonged fever but were finally diagnosed with febrile illness, not KD. Therefore, they did not receive KD treatment because they had no coronary complications on echocardiography and did not meet the laboratory criteria for probable KD. The most frequent symptoms in the FC group were rash, conjunctival injection, and neck lymphadenopathy, and their diagnoses were simple viral exanthema, pharyngo-conjunctival fever caused by adenovirus, and neck lymphadenitis due to virus or bacteria. The causative viruses and bacteria in the FC group were confirmed by serology, virus PCR, and bacterial blood culture tests.

### 2.2. Diagnostic Criteria

The acute phase of KD was defined as the febrile period before intravenous immunoglobulin (IVIG) administration; the subacute phase was defined as the afebrile period 2 days after IVIG administration, and the convalescent phase was defined as 3–4 weeks after the KD event. IVIG resistance was defined as a body temperature of ≥38.0 °C between 36 h and 7 days after the completion of IVIG infusion [6].

The diagnosis of complete KD was based on 5 principal clinical features (conjunctival injection, rash, strawberry tongue or red lips, neck lymphadenopathy, and edema and erythema of the hand and foot). Children were classified as having complete KD (cKD) if they had 4 or 5 of the 5 KD symptoms, with fever for >5 days, and as having incomplete KD (iKD) if they had 2 or 3 of the 5 KD symptoms, with fever for >5 days. We enrolled children with iKD in addition to those with cKD because the differences between iKD and cKD were not significant for all laboratory variables (data not shown).

The laboratory data of each of the 4 KD subgroups were compared with those of the corresponding 4 FC subgroups, and the data of the 4 chronological age subgroups in KD were compared with each other. C-reactive protein (CRP) was not included as a diagnostic criterion in this study.

### 2.3. Z-Value Calculation

We complied with the guidelines for Z-score classification for coronary artery abnormalities as declared in the scientific statement for health professionals from the American Heart Association in 2017 [2]. Coronary artery abnormalities were classified into 2 categories in this study owing to the small number of coronary aneurysm cases: normal coronary artery (Z-score < 2.0) and CALs, including coronary artery dilatation (CAD) and coronary artery aneurysm (CAA) (Z-score ≥ 2.0) [7].

Differential WBC and RBC counts were obtained from the 4 age subgroups in KD and FC. Age-adjusted Z-values were calculated from the mean values and standard deviations (SDs) using age-unadjusted raw values. The neutrophil-to-lymphocyte ratio (NLR) was simply calculated from the neutrophil and lymphocyte counts in each age subgroup.
(1)Z−value = (measure value−mean value)(standard deviation)

The Z-value can correct age-related biases of raw values because different chronological ages have different normal ranges. The SD is needed to calculate the Z-values and is simply derived from the maximal and minimal values by the range rule formula of Taylor.
(2)Standard deviation (SD) ≅ (maxial value−minimal value)4

The mean, maximal, and minimal values for normal WBCs and RBCs according to chronological age have been previously reported, and the mean, SD, and new Z-value of each age were calculated using Equations (1) and (2) [8,9]. The Z-value of leukocyte (Z-leukocyte), Z-value of neutrophil (Z-neutrophil), Z-value of lymphocyte (Z-lymphocyte), Z-value of eosinophil (Z-eosinophil), Z-value of monocyte (Z-monocyte), and Z-value of hemoglobin (Z-hemoglobin) were compared between the KD and FC subjects and among KD age subgroups.

### 2.4. Statistical Analysis

The raw values (only for hemoglobin) and Z-values among the 4 age subgroups were compared in the acute, subacute, and convalescent phases of KD. Statistical analyses were performed using SPSS 20 software (SPSS Inc., Chicago, IL, USA). Data are expressed as the median value (25th percentile Q1, 75th percentile Q3). Continuous variables were compared using the Kruskal–Wallis and Mann–Whitney U test to estimate the significance of incidence. Categorical variables were compared using the Pearson χ^2^ test. The values were compared between infants and non-infants, or among the acute, subacute, and convalescent phases, using a nonparametric test (Mann–Whitney U test). *p* values of <0.05 were considered statistically significant. Receiver operating characteristic (ROC) curve analysis was performed to predict the specific factors and to calculate the optimal cutoff values for differentiating different conditions.

### 2.5. Ethics

The protocol of the present retrospective study was reviewed and approved by the Institutional Review Board of Korea University Guro Hospital (approval no. KUGH10229, 7 March 2019). Informed consent was not applicable for this study owing to the retrospective nature of the analysis.

### 2.6. Echocardiography

A transthoracic 2D echocardiographic examination was performed with sector probes, 5 or 7 MHz and i33 device (Philips, Amsterdam, and Netherland). One physician performed all echocardiographic scans using standard techniques and all the echocardiograms were reviewed by a panel of pediatric cardiologists specialized in echocardiography, according to the Pediatric Heart Network Investigators [7]. There were no significant differences in measurements performed by the same observer for all parameters with intervals of 5–10 min with respect to intra-observer variability (median value differences: *p* > 0.05).

## 3. Results

### 3.1. Patient Characteristics

The number of iKD and the incidences of CAL showed significant differences among the four age groups (Kruskal–Wallis test, respectively, *p* < 0.001), and they showed significant differences between infantile KD and non-infantile iKD (respectively, 36% infants versus 21% non-infants, *p* < 0.001, 18% infants versus 10% non-infants, *p* = 0.004). However, they showed significant differences neither within the infantile KD group (between the EY and TY subgroups) nor within the non-infantile KD group (between the TO and EO subgroups). IVIG resistance in the EY subgroup was significantly lower than that in the TY and TO subgroups (*p* = 0.019 and 0.036 (not shown), respectively) (Table 1).

### 3.2. Binominal Logistic Regressions and Optimal Cutoff Values of the Risk Factors for CAL in KD in Each KD Phase and Age Subgroup

The significant risk factors for CAL in total KD patients were age (decrease), iKD, post-IVIG fever, IVIG resistance, convalescent Z-eosinophil, and subacute platelet (respectively, *p* < 0.05). The significant risk factors for CAL in the EY subgroup were IVIG resistance, acute Z-neutrophil, subacute Z-neutrophil, subacute NLR, and subacute platelet (respectively, *p* < 0.05). The significant risk factors for CAL in the TY subgroup were acute Z-hemoglobin and convalescent Z-hemoglobin (respectively, *p* < 0.05). The significant risk factors for CAL in the TO subgroup were iKD, post-IVIG fever, IVIG resistance, subacute platelet, and subacute Z-monocyte (respectively, *p* < 0.05). The optimal cutoff values of subacute platelet in the total KD patients and TO subgroup for predicting CAL were 506 and 505 (respectively, AUC (area under the curve) = 0.744, 0.749, *p* < 0.001), and that of acute Z-hemoglobin in the TY subgroup for predicting CAL was −2.6 (AUC = 0.766, *p* < 0.001) (Table 2).

There were no significant predictors for CAL in all raw values including hemoglobin.

### 3.3. Comparisons of Hematological Z-Values between KD and FC Subgroups in the Acute Phase

The mean and SD values for each age were newly calculated using Equation (2), and the Z-values were calculated using Equation (1) (the new Table 1 of the published erratum) [10]. The new table has been reconstructed from the textbook Lanzkowsky’s Manual of Pediatric Hematology and Oncology [11].

The parameters of the four KD subgroups were compared with those of the four corresponding FC subgroups in the acute phase. Z-neutrophil, NLR, and CRP of the EY subgroup in KD showed significant differences compared with those of the EY subgroup in FC (respectively, *p* < 0.05). Z-leukocyte, Z-neutrophil, NLR, Z-hemoglobin, platelet, CRP, and ESR (erythrocyte sedimentation rate) of the TY subgroup in KD showed significant differences compared with those of the TY subgroup in FC (respectively, *p* < 0.05). Almost all parameters of the non-infantile subgroups in KD (except Z-lymphocyte in the TO subgroup and Z-monocyte in the EO subgroup) showed significant differences compared with those of the corresponding non-infantile subgroups in FC (Table 3, Figure 1).

Z-neutrophils in the four KD subgroups showed significant increases compared with those in the four FC subgroups (respectively, *p* < 0.05). Z-eosinophils of the TO and EO subgroups in KD showed significant increases compared with those of the corresponding subgroups in FC (respectively, *p* < 0.05). Most Z-hemoglobin in the four KD subgroups showed significant decreases compared with that in the four FC subgroups (respectively, *p* < 0.05, except Z-hemoglobin in the EY subgroup was *p* = 0.850) (Table 3, Figure 1)

### 3.4. Comparisons of Hematological Z-Values among 4 Age Subgroups or among Phases in KD 

Z-neutrophils in the acute and subacute phases showed significant differences among age subgroups (Kruskal–Wallis test, *p* < 0.05), and they showed gradual increases significantly or insignificantly from EY to EO subgroup. Z-neutrophils showed neutrophilia (Z > 0) in the acute phase and neutropenia (Z < 0) in the subacute and convalescent phases in each age subgroup. The NLR changes of four KD subgroups in the acute phase showed the same patterns with those of Z-neutrophils. Z-lymphocytes in all phases showed significant difference among them (Kruskal–Wallis test, *p* < 0.001) and median Z-lymphocytes showed lymphopenia (Z < 0) in all phases and all age subgroups. Z-eosinophils in all phases showed significant differences among them (Kruskal–Wallis test, *p* < 0.05). Median Z-eosinophils showed all eosinopenia (Z < 0) in the acute phase, and they showed most eosinophilia (Z > 0) in the subacute and convalescent phases in each subgroup, except in the convalescent phase of the TO subgroup. Z-hemoglobin in all phases showed significant differences among the four age subgroups in each phase (Kruskal–Wallis test, *p* < 0.001) and median Z-hemoglobin in the TY subgroup showed the significantly lowest negative values compared with those in other age subgroups in each phase (*p* < 0.05, everything not shown in Table 4). Median Z-hemoglobin showed all negative values (Z < 0) in all phases and all age subgroups, and they showed significantly lowest negative values in the subacute phase compared with those in other phases of each age subgroup (*p* < 0.05, everything not shown in Table 4) (Table 4).

### 3.5. ROC Curve to Differentiate True Complete KD Groups and FC Groups in Acute Phase

ROC curve analysis was performed to predict the diagnosis of true complete KD differentiated from FC by calculating the optimal cutoff values in the complete KD patients (excluding incomplete KD) compared with the FC patients. ROC curves of all kinds of Z-values, NLR, platelet CRP, and ESR were performed in the acute phase of KD but the significant ROC curves (AUC > 0.8) were only those of acute NLR and acute CRP. The optimal cutoff value of acute NLR for differentiating complete KD from FC was 1.2 (area under the curve (AUC), 0.821; sensitivity/specificity, 0.796/0.765; *p* < 0.001; Figure 2A), and the cutoff value of acute CRP (mg/L) for differentiating complete KD from FC was 45 (AUC, 0.823; sensitivity/specificity, 0.742/0.765; *p* < 0.001; Figure 2B) only in the TY subgroup. The optimal cutoff values of acute NLR were not obtained in the EY, TO, and EO age subgroups.

## 4. Discussion

Our authors tried to investigate the characteristics of the novel hematological Z-values and to predict the risk of coronary artery lesions through them in four chronological age subgroups of KD.

The hematological Z-value can be a useful standard parameter through which to understand the chronological changes of WBC and RBC counts according to age, although one limitation is that ethnic differences may be present [5]. The age-adjusted Z-values for differential WBC and RBC counts can eliminate age bias because each chronological age has a different normal range (different mean and SD). Because a simple comparison of the age-unadjusted raw values leads to error and the possibility of an incorrect conclusion, our study has used and compared these age-adjusted Z-values.

The risk factors for the clinical outcome of CAA and CAL in KD has been reported to be associated with ages, pre-, post-, and total- IVIG fevers, eosinophil, neutrophil, NLR, and CRP [12]. We attempted to investigate the risk factors associated with CAA in KD but the small number of CAA cases was insufficient to prove statistical relevance. Instead, CAL including CAD and CAA was analyzed with the various parameters. The risk factors for CAL in the total KD patients can be different from those in each subgroup because the specific age subgroup which has a higher number affects more statistical results. However, the significant risk factors for CAL in the specific age subgroup which has a small number can be more statistically important to predict CAL, particularly in the EY subgroup of this study.

Previous studies have revealed that infantile KD is associated with increased CAL due to incomplete presentations compared with non-infantile KD and that infants younger than 6 months showed increased CALs due to incomplete presentations compared with infants older than 6 months [13,14]. Our results showed that younger age and incomplete presentation can be risk factors for CAL (Table 2). Therefore, younger age and incomplete presentation can be independent risk factors for CAL. The number of KD subgroups, particularly in extreme age subgroups, should be reinforced, and their patterns need to be evaluated in the future.

The immune response in earlier life is characterized by immaturity and tolerance of immune function, and immune responses in children with KD younger than 6 months are more suppressed than those in older children with KD [3,15]. Our results showed that younger age can be a risk factor for incomplete presentation (Table 2). The IVIG responsiveness in the EY subgroup was significantly higher than that of the TY and TO subgroups (*p* = 0.019 and 0.036, respectively). The immune response in the EY subgroup may support immune toleration in terms of incomplete presentation and IVIG responsiveness. However, the risk of IVIG resistance for CAL in the EY subgroup was very high (odds ratio 8.433, *p* < 0.001) in the EY subgroup. Therefore, overall IVIG responsiveness in the EY subgroup is high but, partially, IVIG resistance in the EY subgroup is high once coronary complications have occurred. The responsiveness of IVIG in earlier life (< 6 months) may be different according to the presence or absence of the coronary artery complication and further investigation should be performed to elucidate this issue.

We demonstrated that the laboratory characteristics of leukocyte, neutrophil, lymphocyte, and hemoglobin in KD were different from those in FC [5]. Our study showed that the Z-values of the WBC and RBC counts in the < 6 months age subgroup in KD did not show significant differences compared with those in the corresponding age subgroups in FC. Conversely, the > 60 months age subgroup in KD showed significant differences in almost all parameters (increased leukocytosis, neutrophilia, lymphopenia, and anemia; less eosinopenia; increased platelet counts in terms of Z-values, with the exception of monocytes) compared with those in the corresponding age subgroups in FC. The Z-values for ages between 7 and 59 months in KD showed intermediate significant differences only in some parameters (Z-leukocyte and Z-neutrophil) compared with those of the corresponding ages in FCs (Figure 1). These findings support previous results [3,14] and the immune responses of KD in younger ages show more tolerance or less reaction compared with those at older ages during the acute phase in comparison with FC.

During the course of acute-stage KD, the inflammatory cells in coronary arteries consist mainly of infiltrating neutrophils and monocyte/macrophages, and these cells are involved in vascular damage [16,17]. Conversely, the inflammatory cells in lymph nodes consist mainly of lymphocytes, plasma cells, and monocytes/macrophages but include a relatively low number of neutrophils [18,19]. The relationship between immune cells such as neutrophils and monocyte/macrophage and predominant sites such as intravascular, perivascular, and extravascular lesions is not well known. Our results showed an increase in intravascular Z-neutrophil and decrease in Z-monocyte in the acute phase of KD compared with that in FC. The relationship between neutrophils or monocyte/macrophage and infiltrations of intravascular, perivascular, or extravascular lesions should be investigated to elucidate the pathogenesis of KD.

Increased eosinophils were observed during acute, subacute, and convalescent phases and they have been reported as an independent risk factor for CAL during the acute and subacute phases of KD, although they were not age-adjusted Z-values [20,21]. In this study, Z-eosinophil levels in KD subgroups showed significant differences from one another in all phases (Table 4), and the risk of convalescent Z-eosinophil for CAL in total KD patients showed a significant increase (odds ratio 1.288, *p* = 0.001). Elevated eosinophil can have an important role in the development of CALs, although the harmful periods of eosinophil were different according to study design.

Anemia is one of the most common clinical features of KD [22]. Physiological anemia in infancy begins at approximately 3 days of life and typically reaches a nadir at 2–3 months of age [23]. The changes in hemoglobin in KD showed a similar pattern to those in physiological anemia, as they showed the lowest value in the EY subgroup among the four age subgroups in all phases. However, the changes in Z-hemoglobin in KD were different and they showed significantly lowest values in the TY subgroup among the four age subgroups (Table 4). A decrease in age-adjusted Z-hemoglobin was an independent predictor of CAL during the acute and subacute phases [24,25]. In this study, decreases in acute and convalescent Z-hemoglobin in the TY subgroup can increase CAL, and the cutoff value of acute Z-hemoglobin to predict the risk of CAL in the TY subgroup was −2.6. Therefore, anemia based on Z-hemoglobin at 7–12 months of life might be a useful parameter to predict the risk of CAL.

NLR has been reported to be a useful marker for predicting IVIG resistance and CALs and can be a rapid and simple parameter of systemic inflammation in critically ill patients [26,27]. Our results showed that NLR was significantly different between KD and FC in all age subgroups (Table 3) and the risk of subacute NLR for CAL in the EY subgroup showed very high values (Table 2) (odds ratio 23.26, *p* = 0.009). A useful cutoff value of acute NLR to differentiate true complete KD from febrile illness was obtained only in the TY subgroup and it was 1.2 (Figure 2A), but cutoff values to predict CAL during other phases or in the age subgroups were not obtained. Therefore, NLR can be a useful parameter to differentiate true KD from febrile illness and to predict CAL in the specific age subgroups.

Z-neutrophil and Z-monocyte have been reported to be associated with the risk of CAL [26,27,28]. Our study showed that an increase in acute Z-neutrophil, decrease in subacute Z-neutrophil in the EY subgroup, and increase in subacute Z-monocyte in the TO subgroup were associated with the risk of CAL. The changes in neutrophil and monocyte can be useful parameters to predict the risk of CAL in the specific age subgroups.

The vascular complications and responsiveness to IVIG in KD are largely related to the extent of KD inflammation, such as the degrees of concomitant infections and other inflammatory markers [29,30,31]. It is necessary to approach KD by considering the underlying disease and other inflammatory factors along with hematological Z-values.

Many parameters associated with Z-values to predict the risk of CAL in this study were newly discovered, particularly in the EY subgroup. Increases in IVIG resistance, acute Z-neutrophil, subacute NLR, and subacute platelet and a decrease in subacute Z-neutrophil in the EY subgroup can augment the risk of CAL (Table 2). As mentioned above, infants younger than 6 months showed more incomplete presentations and CAL compared with infants older than 6 months [13,14]. Therefore, the risk factors such as IVIG resistance, subacute platelet, subacute NLR, and acute or subacute Z-neutrophil in the EY subgroup can be very useful parameters to predict CAL in young incomplete KD with ambiguous manifestation.

A limitation of this study was that the mean, maximal, and minimal values of WBC and RBC counts for the measurement of Z-values can differ according to age and ethnicity. These mean and SD values according to chronological ages cannot be applied equally among races. Nevertheless, these will certainly help to elucidate the overall changes in laboratory findings.

## 5. Conclusions

Younger age and incomplete presentation in KD can be independent risk factors for CAL. The immune reactions of KD in younger ages are more tolerated or less reactive compared with those at older ages during the acute phase. The immune response in the EY subgroup showed immune tolerance in terms of incomplete presentation and IVIG responsiveness. The responsiveness of IVIG in earlier life (<6 months) may be different according to presence or absence of the coronary artery complication. The risk factors such as IVIG resistance, subacute platelet, subacute NLR, and acute or subacute Z-neutrophil in the EY subgroup can be very useful to predict CAL in young incomplete KD with ambiguous manifestation. Moreover, Z-values such as Z-neutrophil, Z-monocyte, Z-eosinophil, Z-hemoglobin, and NLR can be useful parameters to predict the CAL and to differentiate the true KD from febrile illness in specific age subgroups. Therefore, access to KD needs to be different for each age subgroup, because the hematological trends and CAL rates differ according to age subgroup.

## Figures and Tables

**Figure 1 medicina-56-00466-f001:**
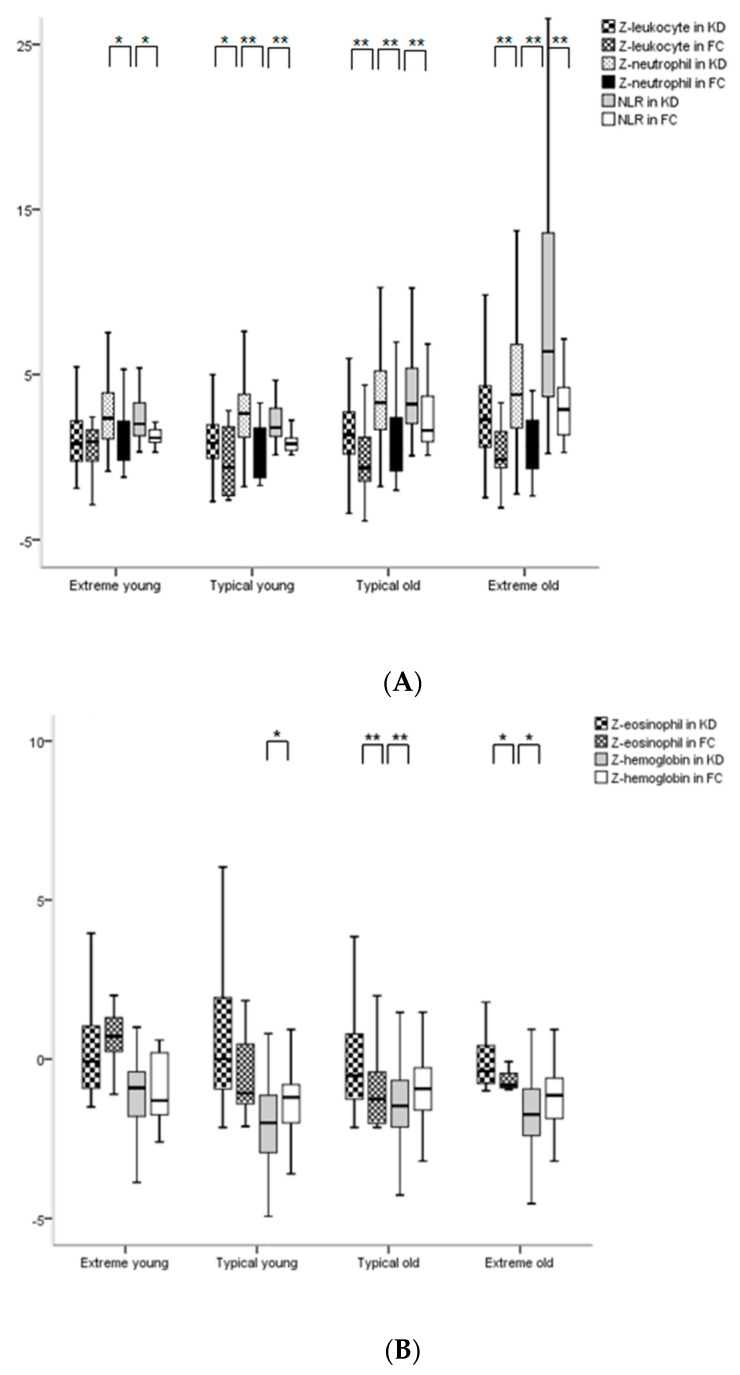
The comparisons of parameters between the 4 KD and FC subgroups according to chronological ages in the acute phase. (**A**) The comparisons of Z-leukocyte, Z-neutrophil, and NLR between two groups. (**B**) The comparisons of Z-eosinophil and Z-hemoglobin between two groups. * means *p* < 0.05, ** means *p* < 0.001. (**C**) The comparisons of platelet between two groups.

**Figure 2 medicina-56-00466-f002:**
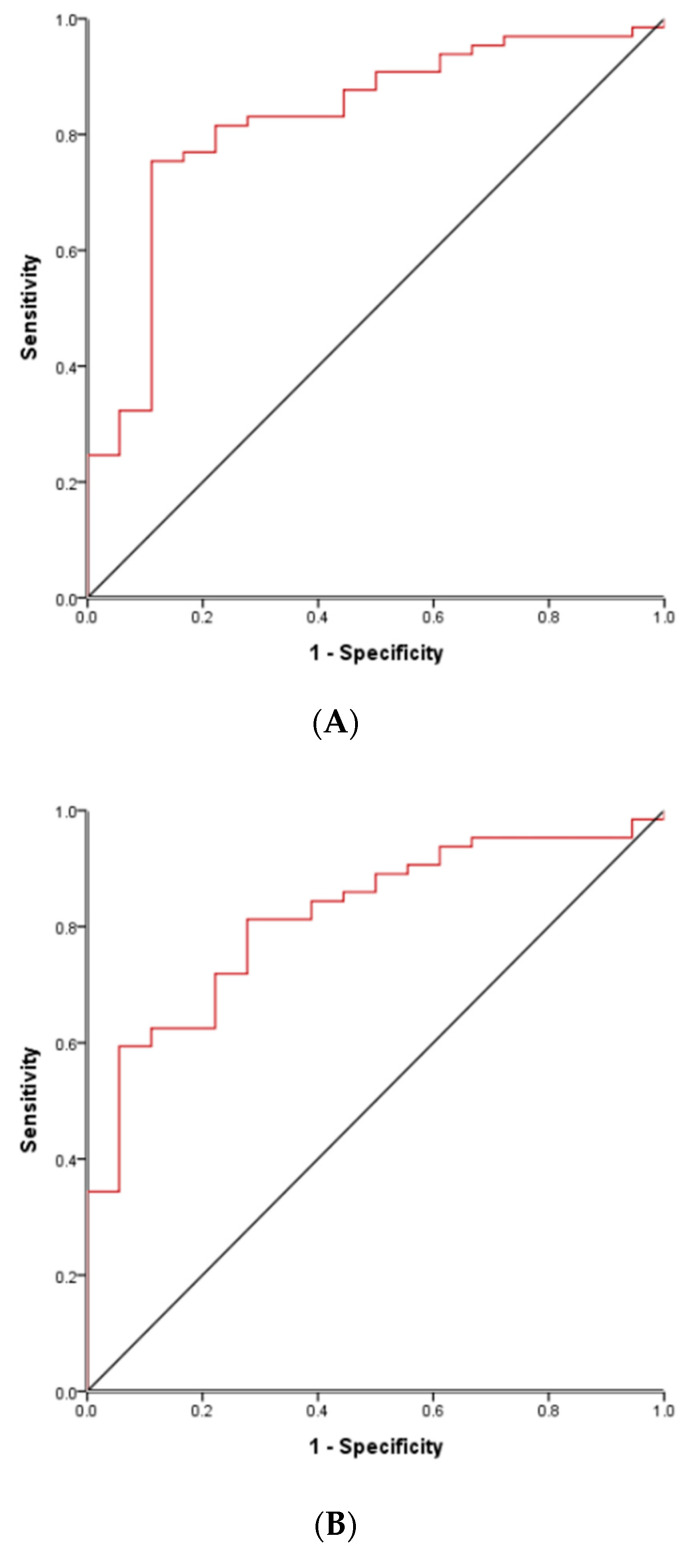
ROC curve of complete KD compared with FC only in the TY subgroup. (**A**) The optimal cutoff value of acute NLR for differentiating complete KD from FC was 1.2 (area under the curve (AUC), 0.821; 95% confidence interval, 0.711–0.931; sensitivity/specificity, 0.796/0.765; *p* < 0.001) in the TY subgroup. (**B**) The optimal cutoff value of acute CRP (mg/L) for differentiating complete KD from FC was 45 (AUC, 0.823; 95% confidence interval, 0.721–0.925; sensitivity/specificity, 0.742/0.765; *p* < 0.001) in the TY subgroup. ROC = receiver operating characteristic, KD = Kawasaki disease, FC = febrile control, TY = typical young, NLR = neutrophil-to-lymphocyte ratio, CRP = C-reactive protein.

**Table 1 medicina-56-00466-t001:** Demographic and clinical characteristics between infantile and non-infantile Kawasaki disease (KD) (Pearson χ^2^ test).

Clinical	Infantile KD (*n* = 223)		Non-Infantile KD (*n* = 681)		*p* Value			
Features	Extreme Young (EY) (Age ≤ 6 Months)	Typical Young (TY) (Age 7–12 Months)	Typical Old (TO) (Age 13–59 Months)	Extreme Old (EO) (Age ≥ 5 Years)	Kruskal-Wallis Test	EY vs. TY	TY vs. TO	TO vs. EO
Numbers	82	141	574	107	-	-	-	-
Age (months)	3 (4, 5)	9 (8, 10)	32 (23, 44)	71 (64, 90)	<0.001	<0.001	<0.001	<0.001
M:F	48:34	77:64	333:241	61:46	0.728	0.936	0.506	0.280
iKD (incomplete KD) (%)	80/223 (36%)		145/681 (21%)		*p* value between infants vs. non-infants: <0.001			
	29/82 (35%)	51/141 (36%)	121/574 (21%)	24/107 (22%)	<0.001	0.687	0.007	0.498
Symptoms	4 (3, 4)	4 (3, 4)	4 (4, 5)	4 (4, 4)	<0.001	0.750	<0.001	0.125
^a^ Total fever	6 (5–7)		6 (7–8)		*p* value between infants vs. non-infants: <0.001			
	6 (5, 7)	6 (5, 8)	7 (6, 8)	7 (6, 8)	<0.001	0.077	0.118	0.271
Pre-IVIG (intravenous immunoglobulin)	5 (4, 5)	5 (4, 6)	5 (4, 6)	5 (4, 7)	0.057	0.542	0.843	0.011
Post-IVIG	1 (0, 2)	2 (1, 3)	2 (1, 3)	2 (1, 3)	<0.001	0.004	0.751	0.112
IVIG rest (%)	22/82 (27%)	60/141 (43%)	207/574 (36%)	34/107 (32%)	0.101	0.019	0.374	0.395
CAL (coronary artery lesion) (%)	40/223 (18%)		70/681 (10%)		*p* value between infants vs. non-infants: 0.004			
	16/82 (20%)	24/141 (17%)	63/574 (11%)	7/107 (7%)	<0.001	0.640	0.061	0.166
CAD (coronary artery dilatation) (%)	11/82 (13%)	18/141 (13%)	46/574 (8%)	2/107 (2%)	0.002	0.890	0.098	0.023
CAA (coronary artery aneurysm) (%)	5/82 (6%)	6/141 (4%)	17/574 (3%)	5/107 (5%)	0.326	0.540	0.427	0.670

“a” Total fever means sum of pre and post IVIG.

**Table 2 medicina-56-00466-t002:** Binominal logistic regressions (A) and cutoff values (B) of the risk factors for coronary artery lesion (CAL) in Kawasaki disease (KD) in each KD phase and age subgroup.

**A.**				
**Subjects**	**Risk Factors for CAL**	**OR**	**95% CI**	***p***
Total patients	Ages	0.984	0.969–0.999	0.041
	Incomplete KD (iKD)	2.079	1.257–3.440	0.004
	^a^ (risk factor of ages for iKD)	(0.991)	(0.984–0.998)	(0.013)
	Post-IVIG fever	1.413	1.274–1.566	<0.001
	IVIG resistance	3.063	1.964–5.034	<0.001
	Convalescent Z-eosinophil	1.288	1.063–1.561	0.010
	Subacute platelet	1.003	1.002–1.005	<0.001
EY subgroup	IVIG resistance	8.433	2.695–26.388	<0.001
	Acute Z-neutrophil	2.636	1.299–5.350	0.007
	Subacute Z-neutrophil	0.127	0.027–0.598	0.009
	Subacute NLR (neutrophil to lymphocyte ratio)	23.26	2.19–247.43	0.009
	Subacute platelet	1.012	1.005–1.020	0.002
TY subgroup	Acute Z-hemoglobin	0.323	0.169–0.618	0.001
	Convalescent Z-hemoglobin	0.223	0.051–0.977	0.047
TO subgroup	iKD	2.870	1.371–6.004	0.005
	Post-IVIG fever	1.374	1.207–1.565	<0.001
	IVIG resistance	3.139	1.495–6.591	0.003
	Subacute platelet	1.004	1.001–1.006	0.002
	Subacute Z-monocyte	1.297	1.028–1.636	0.028
**B.**				
**CAL**	**Cutoff Values**	**Sensitivity/Specificity**	**AUC (95% CI)**	***p* Value**
Subacute platelet	506 k	0.635/0.630	0.744 (0.642–0.846)	<0.001
Subacute platelet in TO	505 k	0.688/0.646	0.749 (0.663–0.834)	<0.001
Acute Z-hemoglobin in TY	−2.6	0.735/0.647	0.766 (0.641–0.892)	0.001

OR = odds ratio, CI = confidence interval, IVIG = intravenous immunoglobulin, EY = extreme young, TY = typical young, TO =typical old, k = 1000. ^a^ The risk factor represents the ages for iKD not for CAL.

**Table 3 medicina-56-00466-t003:** Laboratory characteristics of Kawasaki disease (KD) and febrile control (FC) subgroups according to 4 chronological ages in acute phase.

Acute Variables	Extreme Young (EY) (Age ≤ 6 Months)		*p* Value	Typical Young (TY) (Age 7–12 Months)		*p* Value	Typical Old (TO) (Age 13–59 Months)		*p* Value	Extreme Old (EO) (Age ≥ 5 Years)		*p* Value
	KD (*n* = 82)	FC (*n* = 21)	KD vs. FC	KD (*n* = 141)	FC (*n* = 23)	KD vs. FC	KD (*n* = 574)	FC (*n* = 96)	KD vs. FC	KD (*n* =107)	FC (*n* = 36)	KD vs. FC
Z-Leukocyte	0.7 (−0.2, 2.2)	0.9 (−1.0, 2.2)	0.805	0.9 (−0.1, 2.0)	−0.6 (−2.3, 1.9)	0.001	1.4 (0.2, 2.7)	−0.6 (−1.5, 1.2)	<0.001	2.3 (0.6, 4.5)	−0.1 (−0.6, 1.7)	<0.001
Z-Neutrophil	2.4 (1.1, 3.9)	1.1 (−0.3, 2.5)	0.049	2.6 (1.2, 3.8)	0.2 (−1.3, 1.8)	<0.001	3.3 (1.7, 5.2)	0.4 (−0.8, 2.4)	<0.001	3.8 (1.7, 6.8)	0.9 (−0.7, 2.4)	<0.001
Z-Lymphocyte	−1.1 (−1.6, −3.3)	−0.3 (−0.7, 0.1)	0.058	−1.4 (−0.6, −2.1)	−1.6 (−2.6, 0.3)	0.827	−1.7 (−2.3, −1.1)	−2.0 (−2.5, −1.2)	0.162	−1.4 (−1.8, −0.9)	−1.1 (−1.4, −0.4)	0.018
Z-Eosinophil	−0.1 (−0.9, 1.1)	−0.7 (−1.5, −0.0)	0.162	0.0 (−1.0, 1.9)	−1.2 (−1.4, 0.6)	0.061	−0.5 (−1.3, 0.8)	−1.3 (−2.0, −0.4)	<0.001	−0.5 (−0.8, 0.5)	−0.8 (−0.9, −0.2)	0.006
Z-Monocyte	1.2 (−0.2, 3.1)	2.8 (0.8, 4.4)	0.119	0.8 (−0.4, 1.8)	1.3 (0.2, 2.2)	0.288	0.7 (−0.4, 2.0)	1.2 (−0.1, 3.3)	0.045	1.0 (−0.3, 2.4)	1.1 (0.2, 2.8)	0.299
NLR	2 (1.3, 3.3)	1.2 (0.8, 1.7)	0.009	1.8 (1.3, 3.0)	0.8 (0.4, 1.2)	<0.001	3.2 (2.0, 5.4)	1.6 (0.9, 3.7)	<0.001	6.4 (3.7, 13.6)	2.9 (1.3, 4.8)	<0.001
Z-Hemoglobin	−0.9 (−1.8, −0.4)	−1.3 (−1.9, 0.3)	0.850	−2 (−2.9, −1.1)	−1.2 (−2.0, −0.8)	0.044	−1.5 (−2.1, −0.7)	−0.9 (−1.6, −0.3)	<0.001	−1.7 (−2.4, −0.9)	−1.1 (−1.9, −0.5)	0.038
Platelet (/1000)	383 (332, 461)	433 (224, 540)	0.985	384 (312, 471)	304 (176, 379)	0.001	329 (277, 402)	240 (201, 331)	<0.001	322 (272, 393)	278 (169, 309)	<0.001
CRP (mg/L)	71 (33, 124)	29 (11, 75)	0.026	78 (41, 110)	22 (2, 51)	<0.001	77 (46, 128)	34 (7, 52)	<0.001	89 (56, 135)	39 (23, 80)	<0.001
ESR	56 (41, 73)	68 (29, 79)	0.803	74 (46, 104)	24 (9, 65)	<0.001	80 (58, 99)	46 (26, 72)	<0.001	79 (56, 100)	39 (24, 60)	<0.001

Data shown are medians with interquartile ranges. Z- = Z-value, CRP = C-reactive protein, ESR = erythrocyte sedimentation rate.

**Table 4 medicina-56-00466-t004:** Laboratory characteristics of 4 age subgroups according to chronological age in acute, subacute, and convalescent phases of KD.

Variables	Phases	Infantile KD (*n* = 223)		Non-Infantile KD (*n* = 681)		*p* Values			
		Extreme Young (EY) (≤6 m, *n* = 82)	Typical Young (TY) (7–12 m, *n* = 141)	Typical Old (TO) (13–59 m, *n*= 574)	Extreme Old (EO) (≥5 y, *n* = 107)	Kruskal–Wallis Test	EY vs. TY	TY vs. TO	TO vs. EO
Z-Leukocyte	Acute	0.7 (−0.2, 2.2)	0.9 (−0.1, 2.0)	1.4 (0.2. 2.7)	2.3 (0.6, 4.5)	<0.001	0.899	0.026	<0.001
	Subacute	−0.5 (−1.2, 1.0)	−1.3 (−1.9. −0.2)	−1.0 (−1.5, −0.2)	−0.7 (−1.2, 0.7)	<0.001	0.001	0.033	0.004
	Convalescent	−0.8 (−1.4, −0.3)	−1.0 (−1.3, −0.4)	−0.7 (−1.2, −0.2)	−0.7 (−1.1, −0.2)	0.186	0.734	0.165	0.818
Z-Neutrophil	Acute	2.4 (1.1, 3.9)	2.6 (1.2, 3.8)	3.3 (1.7, 5.2)	3.8 (1.7, 6.8)	<0.001	0.830	0.003	0.119
	Subacute	−0.7 (−1.2, 1.4)	−0.9 (−1.4, −0.3)	−0.6 (−1.1, 0.3)	−0.7 (−1.5, 0.6)	0.023	0.043	0.002	0.367
	Convalescent	−1.0 (−1.4, −0.5)	−0.9 (−1.1, −0.5)	−0.5 (−0.9, 0.1)	−0.9 (−1.3, −0.4)	<0.001	0.192	<0.001	<0.001
Z-Lymphocyte	Acute	−1.1 (−1.6, −0.3)	−1.4 (−2.1, −0.6)	−1.7 (−2.3, −1.1)	−1.4 (−1.8, −0.9)	<0.001	0.012	0.007	0.001
	Subacute	−0.7 (−1.2, −0.2)	−1.5 (−2.0, −0.4)	−1.3 (−1.8, −0.7)	−0.7 (−1.2, −0.2)	<0.001	<0.001	0.469	<0.001
	Convalescent	−0.6 (1.1, −0.2)	−0.8 (−1.4, −0.3)	−0.9 (−1.5, −0.3)	−0.1 (−0.4, 0.3)	<0.001	0.292	0.696	<0.001
Z-Eosinophil	Acute	−0.1 (−0.9, 1.1)	−0.0 (−1.0, 1.9)	−0.5 (−1.3, 0.8)	−0.5 (−0.8, 0.5)	<0.001	0.665	0.002	0.014
	Subacute	1.0 (0.1, 2.1)	0.8 (−0.4, 2.0)	0.5 (−0.4, 1.4)	0.4 (−2.6, 1.1)	0.019	0.199	0.209	0.963
	Convalescent	0.6 (−0.2, 2.0)	0.5 (−0.2, 2.0)	−0.3 (−0.8, 0.7)	0.1 (−0.3, 0.7)	<0.001	0.463	0.002	0.003
Z-Monocyte	Acute	1.2 (−0.2, 3.1)	0.8 (−0.4, 1.8)	0.7 (−0.4, 2.1)	1.0 (−0.3, 2.4)	0.041	0.083	0.853	0.081
	Subacute	0.6 (−0.3, 2.4)	0.2 (−0.6, 1.2)	0.4 (−0.4, 1.3)	0.6 (−0.3, 1.2)	0.116	0.019	0.292	0.482
	Convalescent	0.4 (0.3, 0.7)	0.2 (−0.6, 0.7)	0.1 (−0.4, 0.8)	0.1 (−0.3, 0.8)	0.433	0.541	0.652	0.607
NLR	Acute	2.0 (1.3, 3.3)	1.8 (1.3, 3.0)	3.2 (2.0, 5.4)	6.4 (3.7, 13.6)	<0.001	0.187	<0.001	<0.001
	Subacute	0.4 (0.3, 1.2)	0.4 (0.2, 0.7)	0.7 (0.5, 1.1)	1.2 (0.6, 2.9)	<0.001	0.121	<0.001	<0.001
	Convalescent	0.4 (0.2, 0.5)	0.4 (0.3, 0.5)	0.7 (0.5, 0.9)	0.9 (0.7, 1.2)	<0.001	0.685	<0.001	<0.001
Hemoglobin	Acute	10.9 (10.3, 11.4)	11 (10.3, 11.6)	11.5 (10.9, 12)	12 (11.3, 12.4)	<0.001	0.315	<0.001	<0.001
	Subacute	10.1 (9.6, 10.5)	10.2 (9.6, 11)	10.9 (10.3, 11.5)	11.3 (10.9, 11.9)	<0.001	0.085	<0.001	<0.001
	Convalescent	11 (10.6, 11.7)	11.1 (10.5, 11.6)	11.5 (11.2, 12.1)	12.1 (11.6, 12.7)	<0.001	0.874	<0.001	<0.001
Z-Hemoglobin	Acute	−0.9 (−1.8, −0.4)	−2 (−2.9, −1.1)	−1.5 (−2.1, −0.7)	−1.7 (−2.4. −0.9)	<0.001	<0.001	<0.001	0.127
	Subacute	−1.8 (−2.7, −1.2)	−2.8 (−3.7, −2.0)	−2.1 (−2.9, −1.3)	−2.4 (−3.2, −1.7)	<0.001	<0.001	<0.001	0.070
	Convalescent	−0.8 (−1.4, 0.1)	−1.9 (−2.7, −1.2)	−1.3 (−1.9, −0.5)	−1.3 (−2.1, −0.8)	<0.001	<0.001	<0.001	0.276
Platelet (/1000)	Acute	383 (332, 461)	384 (312, 471)	329 (277, 402)	322 (272, 393)	<0.001	0.988	<0.001	0.443
	Subacute	507 (412, 612)	490 (390, 638)	461 (357, 553)	443 (344, 523)	0.002	0.550	0.026	0.325
	Convalescent	436 (331, 520)	357 (299, 493)	331 (277, 409)	343 (287, 405)	<0.001	0.182	0.028	0.544
CRP (mg/L)	Acute	71 (33, 124)	78 (41, 110)	77 (46, 128)	89 (56, 135)	0.177	0.823	0.354	0.179
	Subacute	27 (10, 50)	16 (7, 33)	17 (8, 32)	25 (10, 54)	0.001	0.012	0.702	0.004
	Convalescent	0.4 (0.2, 1.5)	0.3 (0.2, 1.1)	0.5 (0.3, 2.0)	0.4 (0.3, 1.5)	0.010	0.402	0.002	0.400
ESR	Acute	56 (41, 73)	74 (46, 104)	80 (58, 99)	79 (56, 100)	<0.001	<0.001	0.289	0.608
	Subacute	50 (39, 75)	84 (60, 115)	87 (63, 112)	86 (69, 111)	<0.001	<0.001	0.540	0.909
	Convalescent	10 (5, 20)	16 (8, 30)	23 (14, 41)	28 (12, 53)	<0.001	0.064	0.013	0.569

Data shown are medians with interquartile ranges. Z- = Z-value, CRP = C-reactive protein, ESR = erythrocyte sedimentation rate.

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
