# Peer review of "The Risk Prediction of Coronary Artery Lesions through the Novel Hematological Z-Values in 4 Chronological Age Subgroups of Kawasaki Disease"

_medicina, 2020, doi:10.3390/medicina56090466_

Round 1

Reviewer 1 Report

The paper entitled “The risk prediction of coronary artery lesions through the novel hematological Z-values in 4 chronological age subgroups of Kawasaki disease (KD)” by Ha KS et al is quite interesting to read and well-written, though some sentences may be simplified for clarity.

You could try to give a wider scope to the incipit of the discussion (before entering in the paper objective)?

For instance I suggest to highlight that vascular complications in KD (and also response to IVIG) are largely related to the “extent “of KD inflammation, as shown by different papers (please, cite also Dionne A et al. PLoS One 2018, 13: e0206001; Rigante D et al. Rheumatol Int 2010, 30: 841-6; Abe J, et al. J Allergy Clin Immunol 2008, 122: 1008-13).

At line 267, please unify the two concepts within the sentence “Our result showed younger age can be the risk factor for CAL and incomplete presentation also can be the risk factor for CAL”.

Clarify the sentence at line 273: “immune reactions in KD children younger than 6 months are better tolerated than those in older KD children” and also at line 294 “they show that the immune reactions of KD in younger ages are more tolerated or less reactive compared with those in older ages during the acute phase” (certainly you want to say that tolerance is more frequently observed in children less than 6 months, but the exact sense of the sentence should be clarified).

Attention when you use the verb “to telerate” or the term “toleration” (“tolerance” could be better).

Correct the bracket in the sentence at line 278: “However, the risk of IVIG resistance for CAL in the EY subgroup showed very high (odds ratio 8.433, P < 0.001 in the EY subgroup”.

The list of references has no numbers for many papers.

Author Response

Dear reviewers

Thank you for your detailed comments. I have tried to respond faithfully to what you have said, and if there is any part I answered due to misunderstanding, I would be grateful if you would tell me again.

Review report of Reviewer 1

The paper entitled “The risk prediction of coronary artery lesions through the novel hematological Z-values in 4 chronological age subgroups of Kawasaki disease (KD)” by Ha KS et al is quite interesting to read and well-written, though some sentences may be simplified for clarity.

You could try to give a wider scope to the incipit of the discussion (before entering in the paper objective)?

à Line 256-257: Our authors tried to investigate the characteristics of the novel hematological Z-values and to predict the risk of coronary artery lesions through them in 4 chronological age subgroups of KD.

For instance I suggest to highlight that vascular complications in KD (and also response to IVIG) are largely related to the “extent “of KD inflammation, as shown by different papers (please, cite also Dionne A et al. PLoS One 2018, 13: e0206001; Rigante D et al. Rheumatol Int 2010, 30: 841-6; Abe J, et al. J Allergy Clin Immunol 2008, 122: 1008-13).

à Line 347-350: The vascular complications and responsiveness to IVIG in KD are largely related to the extent of KD inflammation such as the degrees of concomitant infections and other inflammatory markers [29-31]. It is necessary to approach KD by considering the underlying disease and other inflammatory factors along with hematological Z-values.

At line 267, please unify the two concepts within the sentence “Our result showed younger age can be the risk factor for CAL and incomplete presentation also can be the risk factor for CAL”.

à Line 277: Our result showed younger age and incomplete presentation can be the risk factors for CAL (Table 2).

Clarify the sentence at line 273: “immune reactions in KD children younger than 6 months are better tolerated than those in older KD children” and also at line 294 “they show that the immune reactions of KD in younger ages are more tolerated or less reactive compared with those in older ages during the acute phase” (certainly you want to say that tolerance is more frequently observed in children less than 6 months, but the exact sense of the sentence should be clarified).

à Line 282-283: the immune responses in children with KD younger than 6 months are more suppressed than those in older children with KD [3,15].

à Line 303: the immune responses of KD in younger ages show more tolerance or less reaction compared with those in older ages during the acute phase in comparisons with FC.

Attention when you use the verb “to telerate” or the term “toleration” (“tolerance” could be better).

à Line 37-38: The immune response at the age of 6 months or less showed immune tolerance in terms of incomplete presentation and IVIG responsiveness.

à Line 281: The immune response in earlier life is characterized by immaturity and tolerance of immune function,

à Line 303: the immune responses of KD in younger ages show more tolerance or less reaction compared with those in older ages during the acute phase in comparisons with FC.

à Line 366-367: The immune response in the EY subgroup showed immune tolerance in terms of incomplete presentation and IVIG responsiveness.

Correct the bracket in the sentence at line 278: “However, the risk of IVIG resistance for CAL in the EY subgroup showed very high (odds ratio 8.433, P < 0.001 in the EY subgroup”.

à Line 288: However, the risk of IVIG resistance for CAL in the EY subgroup showed very high (odds ratio 8.433, P < 0.001) in the EY subgroup.

The list of references has no numbers for many papers.

àLine 383-457: The lists of references were omitted in the process of editing and our authors corrected the reference numbering.

Reviewer 2 Report

This study describe important issue. It is well designed and written, this appropriate for this journal.

The authors analyzed single-centre experience of coronary artery lesions in age subgroups of Kawasaki diseases. Presented study had a relatively small number of patients and provided data was obtained from single-centre. The manuscript is well written, with potentially interesting topic.

- detailed data for type of echocardiographic equipment should be mentioned (typy and producer of device)
- was echocardiographic data evaluated by 1 physician? CoreLab evaluated recorded examinations?
- any data for hematological diagnosis potentially having impact on WBC RBC PLT levels?

Author Response

Dear reviewers

Thank you for your detailed comments. I have tried to respond faithfully to what you have said, and if there is any part I answered due to misunderstanding, I would be grateful if you would tell me again.

Review report of Reviewer 2

This study describe important issue. It is well designed and written, this appropriate for this journal.

The authors analyzed single-centre experience of coronary artery lesions in age subgroups of Kawasaki diseases. Presented study had a relatively small number of patients and provided data was obtained from single-centre. The manuscript is well written, with potentially interesting topic.

- detailed data for type of echocardiographic equipment should be mentioned (typy and producer of device)

à Line 140: A transthoracic 2D echocardiographic examination was performed with sector probes, 5 or 7 MHz and i33 device (Philips, Amsterdam, and Netherland).

- was echocardiographic data evaluated by 1 physician? CoreLab evaluated recorded examinations?

à Line 141: One physician performed all echocardiographic scans using standard techniques and all the echocardiograms were reviewed by a panel of pediatric cardiologists specialized in echocardiography, according to the Pediatric Heart Network Investigators [7]. There were no significant differences in measurements performed by the same observer for all parameters with interval of 5–10 minutes in aspect of intra-observer variability (median value differences: P > 0.05).

- any data for hematological diagnosis potentially having impact on WBC RBC PLT levels?

à The hematologic biomarker associated with WBC levels in Kawasaki disease is said to be related to CRP. In our study, CRP showed meaningful results, but this has already been studied in many other papers, so this study did not discuss it.